# Telemedicine-Based Specialized Care Improves the Outcome of Anticoagulated Individuals with Venous Thromboembolism—Results from the thrombEVAL Study

**DOI:** 10.3390/jcm9103281

**Published:** 2020-10-13

**Authors:** Karsten Keller, Sebastian Göbel, Vincent ten Cate, Marina Panova-Noeva, Lisa Eggebrecht, Markus Nagler, Meike Coldewey, Maike Foebel, Christoph Bickel, Michael Lauterbach, Christine Espinola-Klein, Karl J. Lackner, Hugo ten Cate, Thomas Münzel, Philipp S. Wild, Jürgen H. Prochaska

**Affiliations:** 1Center for Thrombosis and Hemostasis (CTH), University Medical Center of the Johannes Gutenberg-University Mainz, 55131 Mainz, Germany; karsten.keller@unimedizin-mainz.de (K.K.); vincent.tenCate@unimedizin-mainz.de (V.t.C.); Marina.Panova-Noeva@unimedizin-mainz.de (M.P.-N.); lisa.eggebrecht@uni-mainz.de (L.E.); markus.nagler@unimedizin-mainz.de (M.N.); meiko@googlemail.de (M.C.); maike@arenaweb.de (M.F.); h.tencate@maastrichtuniversity.nl (H.t.C.); tmuenzel@uni-mainz.de (T.M.); philipp.wild@unimedizin-mainz.de (P.S.W.); 2Department of Cardiology, Cardiology I, University Medical Center of the Johannes Gutenberg-University Mainz, 55131 Mainz, Germany; sebastian.goebel@unimedizin-mainz.de (S.G.); christine.espinola-klein@unimedizin-mainz.de (C.E.-K.); 3German Center for Cardiovascular Research (DZHK), Partner Site Rhine Main, 55131 Mainz, Germany; Karl.Lackner@unimedizin-mainz.de; 4Preventive Cardiology and Preventive Medicine—Center of Cardiology, University Medical Center of the Johannes Gutenberg-University Mainz, 55131 Mainz, Germany; 5Department of Medicine I, Federal Armed Forces Central Hospital Koblenz, 56072 Koblenz, Germany; christophbickel@bundeswehr.org; 6Department of Medicine 3, Barmherzige Brüder Hospital, 54292 Trier, Germany; lauterbach@uni-mainz.de; 7Institute of Clinical Chemistry and Laboratory Medicine, University Medical Center of the Johannes Gutenberg-University Mainz, 55131 Mainz, Germany; 8Thrombosis Center Maastricht, Cardiovascular Research Institute Maastricht and Maastricht University Medical Center, 6229HX Maastricht, The Netherlands

**Keywords:** venous thromboembolism, oral anticoagulation therapy, vitamin K antagonists, coagulation service, e-health

## Abstract

Venous thromboembolism (VTE) is a life-threatening disease with risk of recurrence. Oral anticoagulation (OAC) with vitamin K antagonists (VKA) is effective to prevent thromboembolic recurrence. We aimed to investigate the quality of OAC of VTE patients in regular medical care (RMC) compared to a telemedicine-based coagulation service (CS). The thrombEVAL study (NCT01809015) is a prospective, multi-center study to investigate OAC treatment (recruitment: January 2011–March 2013). Patients were evaluated using clinical visits, computer-assisted personal interviews, self-reported data and laboratory measurements according to standard operating procedures. Overall, 360 patients with VTE from RMC and 254 from CS were included. Time in therapeutic range (TTR) was higher in CS compared to RMC (76.9% (interquartile range [IQR] 63.2–87.1%) vs. 69.5% (52.3–85.6%), *p* < 0.001). Crude rate of thromboembolic events (rate ratio [RR] 11.33 (95% confidence interval [CI] 1.85–465.26), *p* = 0.0015), clinically relevant bleeding (RR 6.80 (2.52–25.76), *p* < 0.001), hospitalizations (RR 2.54 (1.94–3.39), *p* < 0.001) and mortality under OAC (RR 5.89 (2.40–18.75), *p* < 0.001) were consistently higher in RMC compared with CS. Patients in RMC had higher risk for primary outcome (clinically relevant bleedings, thromboembolic events and mortality, hazard ratio [HR] 5.39 (95%CI 2.81–10.33), *p* < 0.0001), mortality (HR 5.54 (2.22–13.84), *p* = 0.00025), thromboembolic events (HR 6.41 (1.51–27.24), *p* = 0.012), clinically relevant bleeding (HR 5.31 (1.89–14.89), *p* = 0.0015) and hospitalization (HR 1.84 (1.34–2.55), *p* = 0.0002). Benefits of CS care were still observed after adjusting for comorbidities and TTR. In conclusion, anticoagulation quality and outcome of VTE patients undergoing VKA treatment was significantly better in CS than in RMC. Patients treated in CS had lower rates of adverse events, hospitalizations and lower mortality. CS was prognostically relevant, beyond providing advantages of improved international ratio (INR) monitoring.

## 1. Introduction

Venous thromboembolism (VTE) is a major cause of morbidity and mortality with a high risk of recurrence [1,2,3,4]. Cumulative incidences of recurrence in unprovoked VTE are 16%, 25% and 36% after 2, 5 and 10 years, respectively [5]. Oral anticoagulation (OAC) with vitamin K antagonists (VKA) is effective to prevent thromboembolic complications in various clinical settings [6,7,8,9]. Anticoagulation therapy with VKA reduces the risk for recurrent VTE by 90% [7,8] and for fatal VTE by 92% [10,11]. For VTE events, American and European guidelines recommend oral anticoagulation of at least 3 months with a VKA or alternatively, a direct-acting oral anticoagulant (direct-acting non-VKA anticoagulant, DOAC). Extended anticoagulation can be necessary for secondary prevention depending on factors increasing the risk of recurrence of VTE [3,12].

Although the initiation of DOAC-based therapy has significantly increased over time, VKA are still regularly applied to VTE patients in clinical routine, especially to patients who are not eligible for DOAC treatment [3,4,12,13].

The benefit of VKA therapy highly depends on the quality of OAC control [14,15,16,17]. After VTE events, an international normalized ratio (INR) therapeutic range of 2.0–3.0 is recommended [3,12,18,19]. Adjustment of VKA therapy is challenging due to comorbidities, polypharmacy, diet and other factors influencing precise adjustment within the narrow target range. Against this background, different approaches have been proposed to optimize management of VKA treatment [14]. Although a telemedicine-based, specialized coagulation service was recently reported to improve the outcome of patients receiving VKA [20,21,22,23,24], the current recommendations of the American College of Clinical Pharmacy (ACCP) do not support anticoagulation management services due to the heterogeneity of study results [25]. In addition, telemedicine-based coagulation services are promising approaches to also address other fields of health care including minimization of community transmission of infectious diseases, reduction of health care costs and remote management of individuals with specific diseases (e.g., patients with heart failure).

Thus, we aimed to investigate the quality and subsequent clinical outcome of VTE patients receiving phenprocoumon by comparing standard care by physicians to management by a central telemedicine-based coagulation service in a real-life setting.

## 2. Methods

### 2.1. Study Design

The thrombEVAL study program (registered at: http://clinicaltrials.gov, unique identifier: NCT01809015) is an investigator-initiated, prospective, multi-center study program evaluating the potential improvement of quality of anticoagulation therapy and clinical outcome by a specialized coagulation service. The rationale and design of the thrombEVAL study has been described in detail previously [26]. Briefly, the project comprised two observational cohorts: a multi-center cohort study with 21 study centers in regular medical care (RMC) and a single-center, multi-local cohort study in a specialized, telemedicine-based coagulation service (CS) (Appendix A in the Appendix A). Detailed descriptions of both cohorts are given in the Appendix A. Study participants of both cohorts were enrolled between January 2011 and March 2013. Firstly, patients of the RMC cohort were enrolled during their hospitalization in 21 different hospitals. VTE patients with an indication for anticoagulant treatment of at least 4 months duration were asked to participate in our observational study. Exclusion criteria comprised age < 18 years, withdrawal of prior given consent, contraindications to VKA treatment (e.g., pregnancy or known hypersensitivity) or participation in other clinical trials. All patients who gave written informed consent, not fulfilling these exclusion criteria, were included in the RMC group without any further selection (Appendix A in the Appendix A). Secondly, regarding the CS cohort, VTE patients with an indication of at least 3 months of anticoagulant VKA treatment without any of the mentioned exclusion criteria were included in the CS group after giving written informed consent (Appendix A in the Appendix A). These patients, who were included in the CS, were on the one hand hospitalized or ambulatory-treated patients at the University Medical Center Mainz (Germany), or on the other hand, patients who were referred from their physicians to the CS for the management of the anticoagulant treatment. Therefore, the CS cohort comprises VTE patients, who were anticoagulation-naïve (new VTE diagnosis), as well as VTE patients, who were already under anticoagulation treatment, switching from RMC to the anticoagulant treatment managed by the CS. Patients with short-term (e.g., first VTE event without VTE risk factors) as well as long-term (e.g., recurrent VTE) indications for OAC-treatment were enrolled. Study coordination and database management as well as primary analysis were independently performed by the Center for Thrombosis and Hemostasis (Mainz, Germany). Study monitoring was performed by independent institution (Interdisciplinary Center for Clinical Trials (IZKS), Germany). All participants gave written informed consent prior to study enrollment. All procedures followed the principles of good clinical practice as well as Strengthening the Reporting of Observational studies in Epidemiology (STROBE) guidelines and the Declaration of Helsinki. Local ethics boards and the local data safety commissioner approved the study protocol (medical association Rhine-Hessen, Germany; reference no. 837.407.10.7415/7416). Clinical Trial Registration: URL: http://www.clinicaltrials.gov; unique identifier NCT01809015

### 2.2. Assessment of Study Data

Participants of both cohorts received identical, detailed clinical assessment at study inclusion and data obtained from clinical visits, medical records and laboratory measurements were integrated into an electronic patient file according to standard operating procedures. Annual standardized computer-assisted telephone interviews were conducted by trained staff for outcome assessment. All information on study endpoints was validated on the basis of medical records and adjudicated by an independent review panel. In addition, electronic database systems of hospitals’ records were queried for the occurrence of outcome events. We included an intra-individual comparison regarding the quality of oral anticoagulation of patients treated firstly in RMC and afterwards in coagulation service. To obtain this intra-individual comparison, data of patients, who were previously treated in RMC and afterwards included in the CS, were separated in data regarding the treatment before inclusion in the CS and data of the period during the management in the CS [14]. The data, which belong to the period of the anticoagulant treatment before inclusion in the CS, were incorporated in the RMC cohort [14]. This approach enabled us to analyze an intra-individual comparison of these patients. Regarding the quality of anticoagulation control, INR values were obtained from anticoagulation documentation in RMC (e.g., from the anticoagulation pass) for a maximum period of 3 years and electronic patient files in the CS were queried for a maximum duration of 2 years.

### 2.3. Definition of Outcome Parameters

All of the reported adverse events were under OAC-treatment with inclusion of short bridging episodes or short treatment breaks. The primary study outcome was defined as the composite of stroke/transient ischemic attack (TIA), systemic embolism, recurrent VTE event (pulmonary embolism [PE] or deep vein thrombosis (DVT)), myocardial infarction, major and clinically relevant non-major bleeding and all-cause mortality. Stroke was defined as an acute onset of a neurologic deficit of presumed vascular origin lasting for ≥24 h or resulting in death, whereas TIA comprised a neurologic deficit lasting <24 h. Systemic embolisms were defined as acute vascular occlusions of extremities or of any other organ. Thromboembolic events consist of stroke/TIA, systemic embolism, recurrent VTE (PE and/or DVT) and myocardial infarction. Major bleeding was defined as (i) a bleeding event with reduction in hemoglobin level of at least 2.0 g/L, or (ii) a bleeding event leading to transfusion of at least 2 units of blood or packed cells, or (iii) a symptomatic bleeding in a critical area or organ such as retroperitoneal bleeding, intra-articular bleeding or pericardial bleeding. Clinically relevant non-major bleeding was defined as a bleeding event requiring medical attention in either an ambulatory or clinical setting. Clinically relevant bleeding represents the composite endpoint of major bleeding and clinically relevant non-major bleeding.

### 2.4. Statistical Analysis

For the current analysis, data were restricted to individuals with VTE (PE and/or DVT) in both study arm cohorts. For descriptive analysis of baseline characteristics, dichotomous variables were presented by absolute and relative frequencies and differences assessed with Fisher’s Exact Test. Continuous variables were described using the mean and standard deviation (SD), and in case of a markedly non-normal distribution, by the median and the 25th and 75th percentiles (inter-quartile range, IQR). Statistical comparisons for continuous variables were made using the Mann–Whitney *U*-test in case of non-normal distributions; otherwise, an unpaired *t*-test was applied.

In order to evaluate the quality of anticoagulation therapy, TTR was calculated by the linear interpolation method according to Rosendaal et al. [17].

Classification as “patients with stable anticoagulation control” (within individual therapeutic target range) required VKA treatment of at least 28 days with three consecutive INR measurements within the therapeutic range [14]. The profile of time outside the therapeutic range was computed and subdivided into time above and below target range. TTR and time outside therapeutic range were expressed as median values with IQR, and the corrected Z-test was used to compare groups regarding TTR, while the variability of time spent outside the target range was compared using the one-sided Ansari–Bradley test.

For RMC and CS, all adverse events under OAC therapy within the first 2 years after study enrollment were taken into account. Rate ratios (RR) were calculated with corresponding 95% confidence intervals (CI) and *p*-values for clinical outcome, and they were compared using the binomial tests. For prospective follow-up, Cox proportional hazards regression was used to estimate the differences in time to event between cohorts in multivariable models with adjustment for potential confounders (adjusted for age, sex, Charlson Index and TTR). Results were presented as hazard ratios (HR) and 95% CI.

Propensity score weighting was additionally applied in outcome analyses to account for differences in clinical characteristics between cohorts. Standardized mortality ratio (SMR) weighting was used, i.e., the control group (RMC) was unaltered, while the treatment group (CS) was weighted to match the distribution of characteristics of the control group. In technical terms, this means the coefficient estimates in the propensity score-weighted regressions represent the average treatment effect (ATE) for the controls, obviating the need to further adjust for confounding. In addition, we performed a competing risks analysis in order to confirm the Cox regression results, as described above.

We compared the included VTE patients in terms of incident vs. recurrent VTE, PE vs. DVT as well as short- vs. long-term indication for OAC treatment. In addition, subgroups defined a priori were screened for potential interactions. Relative risk of clinically relevant bleeding as well as primary outcome for patients in CS in comparison to those in the RMC, according to different subgroups, were analyzed with the Wald test.

All statistical tests were two-sided. *p*-values < 0.05 were considered to indicate statistical significance. Statistical analyses were performed with R (https://www.R-project.org/; R Foundation for Statistical Computing, Vienna, Austria), version 3.4.3.

## 3. Results

### 3.1. Analysis of Key Data and Patient Characteristics

In the total sample, 614 individuals with VTE were investigated: 360 (58.6%) subjects from RMC and 254 (41.4%) subjects from the CS. A total of 3146.2 treatment years and 11,929 INR measurements were available for TTR analysis. In both cohorts, phenprocoumon was the predominantly administered VKA, being used by 98% of the participants.

VTE patients in RMC were older and showed a pronounced profile of comorbidities (i.e., higher Charlson Index: 5.67 ± 2.56 vs. 4.42 ± 2.71, *p* < 0.0001) compared to CS patients. Arterial hypertension was the most common cardiovascular risk factor in both groups. Notably, cardiovascular disease such as coronary artery disease, history of myocardial infarction, heart failure and peripheral artery disease were more frequently observed in individuals in RMC than in those in CS (Table 1). Calculation of the TTR was possible in 256 of 360 RMC patients (71.1%) and in 240 of 254 of the CS patients (94.5%).

While CS patients were more often in employment, RMC patients more frequently had children. Educational level, the rate of partnerships as well as the proportion of participants living in nursing homes were comparable between both groups.

### 3.2. Quality of Anticoagulation Therapy

TTR was significantly higher in CS than in RMC patients (76.9% (IQR 63.2–87.1%) vs. 69.5% (52.3–85.6%), *p* < 0.001) (Table 1, Appendix A in the Appendix A).The proportion of VTE patients with stable anticoagulation control was also larger in CS compared to RMC (86.2% vs. 75.4%, *p* = 0.003) accompanied by higher median TTR values for those patients in CS (79.5% (IQR 68.2–87.6%) vs. 74.2% (61.6–87.1%) in RMC, *p* = 0.015) (Table 1, Appendix A in the Appendix A). Additionally, the TTR of CS patients, who were previously treated in RMC, was significantly higher after the shift from RMC to CS management (before: 68.9% (IQR 46.5–85.2%) vs. after: 77.8% (67.6–89.5%), *p* = 0.021). In contrast, TTR of patients who measured their INR values on their own did not differ significantly between both groups (RMC: 88.2% (75.6–95.3%) vs. CS: 87.4% (64.9–97.6%), *p* = 0.80). The profile of time spent outside the therapeutic range was more balanced in CS than in RMC, with less time spent below the targeted INR range (Table 1, Appendix A in the Appendix A).

### 3.3. Comparison of Clinical Outcome between Regular Medical Care and Coagulation Service

Mean follow-up times for RMC and CS were 19.9 ± 7.4 and 12.8 ± 7.3 months, respectively. Rates of all assessed adverse events were consistently higher in RMC than in CS patients. Primary outcome events occurred 6.8-fold more often in RMC than in CS. OAC-specific outcomes, comprising thromboembolic events as the efficiency outcome and clinically relevant bleeding as the safety outcome, were respectively 11.3-fold and 6.8-fold more common in RMC compared to CS. OAC-unspecific outcomes such as all-cause mortality and hospitalizations were respectively 5.9-fold and 2.5-fold as frequent in RMC patients relative to CS patients (Table 2).

Unadjusted hazard ratios consistently demonstrated a significantly higher hazard for RMC patients to develop adverse events as indicated by the primary outcome (HR 5.87 (95% CI 3.06–11.26), *p* < 0.0001), all-cause mortality (HR 6.23 (95% CI 2.49–15.56), *p* < 0.0001), clinically relevant bleeding (HR 5.69 (95% CI 2.03–15.93), *p* < 0.001), thromboembolic events (HR 6.85 (95% CI 1.62–29.04), *p* = 0.0090) and hospitalizations (HR 1.93 (95% CI 1.40–2.66), *p* < 0.0001) compared to CS patients (Appendix A in the Appendix A). Since the patient characteristics of the RMC and CS groups were different (as shown in Table 1), we adjusted the Cox regression models for age and sex as well as age, sex and Charlson Index, and furthermore for age, sex, Charlson Index and TTR in order to test the widespread independence of these parameters. These adjusted Cox regression models revealed stable results regarding the benefit of CS on the mentioned adverse events (Table 3).

To further evaluate the robustness of these findings, propensity score-weighted Cox regression was additionally performed. In brief, the previous findings were supported. Subjects treated in RMC showed a higher risk of developing adverse outcomes in comparison to CS: primary outcome (HR 5.63 (95% CI 3.33–9.52), *p* < 0.0001), all-cause mortality (HR 6.59 (95% CI 3.05–14.22), *p* < 0.0001), clinically relevant bleeding (HR 5.06 (95% CI 2.25–11.35), *p* < 0.0001), thromboembolic events (HR 6.24 (95% CI 2.02–19.28), *p* = 0.0015) and hospitalizations (HR 1.58 (95% CI 1.22–2.04), *p* < 0.001) (Figure 1). The accuracy of propensity weighting was shown in Appendix A in the Appendix A.

Finally, the advantages of the CS in comparison to RMC were confirmed in the competing risks analysis (Appendix A in the Appendix A).

### 3.4. Subgroup Analyses

In Figure 2, sensitivity analyses for pre-defined subgroups are shown for the outcome clinically relevant bleeding. Unexpectedly, no significant interactions were identified in terms of better treatment adjustment (TTR > 70% vs. TTR ≤ 70%), co-therapy with acetylsalicylic acid and renal disease.

The key findings of the subgroup analyses are (i) that no significant interactions were found in clinically relevant bleeding (Figure 2) and primary outcome (Appendix A in the Appendix A) regarding treatment in CS and RMC, (ii) treatment benefits driven by CS were independent of incident vs. recurrent status and (iii) that treatment benefits were independent of short-term vs. long-term indication of OAC.

In patients aged ≥ 80 years and those who had renal insufficiency, treatment in CS revealed significant benefits with respect to the primary outcome compared to treatment by RMC (HR 4.17 (95%CI 1.72–10.00), *p* = 0.0016). Higher rates of primary outcome events were detected in RMC patients in comparison to CS patients in the subgroup of PE patients (with or without concomitant DVT) (HR 6.65 (95%CI 2.32–19.07), *p* < 0.001) as well as in the subgroup with DVT patients (HR 4.10 (95%CI 1.70–9.87), *p* = 0.0017) respectively, after adjustment for age, sex, TTR and Charlson Index. VTE patients with a history of recurrent VTE had a higher risk of primary outcome events in both CS and RMC compared to those patients with just a single VTE event in their medical history during the follow-up period (Figure 3).

Patients with single (HR 4.45 (95%CI 2.20–8.96), *p* < 0.0001) as well as patients with recurrent (HR 5.45 (95%CI 2.79–10.63), *p* < 0.0001) VTE events showed higher primary outcome rates in RMC compared to CS (adjusted for age, sex and Charlson Index). This finding was supported by the results of the Cox regression analysis, demonstrating that management of OAC therapy in RMC was independently associated with higher incidence of adverse events regarding the primary outcome (HR 4.87 (95%CI 2.53–9.37), *p* < 0.0001) compared to OAC management by the telemedicine-based CS (independent of age, sex, Charlson Index and the differentiation between single and recurrent VTE). Moreover, VTE patients with a single VTE event and without thrombophilia had a six-fold higher hazard of experiencing the primary outcome in RMC after 6 months (HR 6.12 (95%CI 1.40–26.71), *p* = 0.016) and VTE patients with recurrent events or single events accompanied by existing thrombophilia showed consistently higher risk for the primary outcome in the 12-month follow-up (HR 3.57 (95%CI 1.51–8.42), *p* = 0.0036) in RMC compared to CS, respectively (Figure 4, Appendix A in the Appendix A).

## 4. Discussion

To the best of our knowledge, the thrombEVAL study program is the first real-world study to compare the quality and outcome of phenprocoumon-based VKA therapy in VTE patients between RMC and a telemedicine-based CS. The main study findings can be summarized as follows: (i) CS substantially improved OAC treatment quality according to surrogate parameter TTR compared to RMC, (ii) VTE patients treated in CS had lower rates of anticoagulation-specific adverse events, hospitalizations and all-cause mortality, (iii) the improved outcome of the CS cohort was independent of treatment duration and frequency of VTE events and (iv) the ameliorated outcome was not fully explained by an improvement in TTR, highlighting the value of standardized supportive care for anticoagulated VTE patients.

The TTR obtained by CS in our study was significantly higher than in RMC (76.9% vs. 69.5%, *p* < 0.001) and higher than the median TTR values of the control groups in the large registration trials for the DOACs in VTE patients [1,7,27,28] as well as of two meta-analyses, both with >25,000 VKA-treated VTE patients (56.7% [20] and 60.0% [29]). This might in part be explained by a VKA class-effect, because phenprocoumon, as a long-acting VKA drug, was in some studies associated with higher TTR values than shorter-acting VKA drugs [30]. In line with our results, several studies reported a better anticoagulation quality achieved by CS compared to RMC [21,22]. Focusing on VTE patients, the AuriculA study [31] reported similar findings to those of the present study (median TTR of 73.6% achieved by CS) [31]. By contrast, other studies did not report similarly high percentages of TTR, reporting median TTR values below 70% [21,22,32].

Extending beyond these reported results, in this study, CS patients had a more balanced profile of time spent outside the therapeutic range. In RMC and likewise in all registration trials for DOACs in VTE patients, the time spent below the therapeutic range was substantial [1,7,27,33], which entails an elevated risk of thromboembolic events and is therefore undesirable in VTE patients [34,35].

Although in the ARISTOTLE and RELY trials, no benefits were observed for improved Warfarin management compared to using fixed-dose apixaban or dabigatran with respect to the incidence of stroke, systemic embolism and intracranial bleeding in atrial fibrillation (AF) patients [36,37], the sub-analysis of the RELY study in AF patients demonstrated that dabigatran (at an administered dose of 150 mg) was no longer superior in terms of reduced risk of non-hemorrhagic stroke and systemic embolism when compared with warfarin-treated patients with TTR values > 72.6% [36]. It remains speculative whether this trend would also be observed for patients with higher adjusted mean TTR, as in this present study. In line with this finding, Wan et al. [16] reported that in AF patients, a 7% improvement in TTR led to a reduced incidence of major hemorrhage, and a 12% improvement in TTR was associated with a reduction of one thromboembolic complication per 100 patient-years, respectively [16]. Further, Gallagher et al. [38] showed that patients with higher TTR had a lower risk of recurrent VTE events. Thus, it is of eminent importance to reach a high quality of OAC therapy with high TTR values to reduce the risk of adverse outcomes.

In accordance with the literature, CS management significantly reduced adverse events such as bleeding and thromboembolic events, death and hospitalizations [6,21,23]. In the present study, treatment in CS was associated with a 0.15-fold risk for the primary outcome, 0.09-fold risk for thromboembolic events, 0.15-fold risk for relevant bleeding events, 0.17-fold risk for all-cause mortality and a 0.39-fold risk for hospitalizations compared with RMC during follow-up. These benefits were similar to results reported for integrated care versus primary care in other settings, like atrial fibrillation [39]. We previously reported significantly improved clinical outcomes for the whole thrombEVAL study sample, comprising all patients with various indications of OAC treatment [40]. Nevertheless, there are some general differences regarding OAC treatment of VTE patients in comparison to those patients with OAC treatment for AF or mechanical valves: in contrast to patients with OAC treatment due to AF or mechanical valves, whose OAC indications in the majority of cases are focused on primary prevention of ischemic stroke or valve complications, the treatment of VTE represents the combination of therapy for acute PE or DVT to solve thrombus or embolus and of secondary prevention to avoid recurrent VTE events. The fact that VKA management requires optimal anticoagulation quality to minimize risks including death due to recurrent PE provides a strong rationale for achieving optimal TTR in patients that are still treated with VKA [1,2,3,5,18,34,35,41,42].

Remarkably, no significant interactions were detected in the a priori defined subgroups. In brief, we observed comparable treatment benefits due to CS in the subgroups of patients with renal insufficiency as well as in patients aged ≥ 80 years. This finding is of outstanding interest, since these patients might be candidates alternatively treated with reduced doses of DOACs. However, studies have demonstrated that reduced doses of DOACs are less effective than full dosing, and that reduced dose DOACs might even no longer be superior to VKA treatment [41,43,44]. Hence, CS-guided VKA management may be considered a suitable alternative.

Although treatment in CS was associated with significantly better TTR, the observed reduction in adverse events was only in part attributable to better dosing. After adjustment for TTR in the regression analysis, the benefits of CS treatment remained stable, indicating additional effects related to supportive care of CS. These additional positive effects (beyond better dosing) of the CS might primarily be attributed to better and closer supervision of the anticoagulated patients, including better management of particular situations such as preparation for surgeries, perioperative OAC management or infections. Furthermore, the achieved outcome benefit by CS was not dependent on treatment duration or single vs. recurrent status of the VTE event. Notably, the achieved risk reductions regarding bleeding events as well as thromboembolic events of CS versus RMC were higher than the reported reductions achieved by DOAC treatment compared to warfarin in VTE patients [1,7,27,28]. Nevertheless, it has to be mentioned that patients, who were enrolled for CS, might be more aware and be more motivated to achieve an optimized/better VKA management. Therefore, this difference in motivation and awareness might have resulted in a better adherence to the treatment recommendations, an improved TTR and a lower incidence of adverse events and treatment complications in the patients of the CS.

Although it can be expected that DOACs will replace VKA treatment as standard therapy in VTE, a recently published nationwide registry study reported that in real-world settings, VKA treatment in VTE patients was accompanied by a similar risk for recurrent VTE and bleeding compared to DOACs [45], and it has to be kept in mind that DOAC treatment is not suitable for all VTE patients. As recommended, VKA treatment may be preferred in selected patients, including individuals with impaired renal function (especially those with creatinine clearance < 30 mL/min/1.73m^2^), dyspepsia and poor compliance [12,18]. Although the concept of CS treatment was primarily developed for VKA treatment, our data suggest that CS might also be beneficial in OAC patients treated with DOACs. This beneficial and positive effect might be primarily explained by the supportive care of the CS in all OAC patients [3,24,44,46,47]. Thus, many CS in the United States have adapted their services to also provide care for patients taking DOACs [24,46].

These benefits may be explained by the multi-factorial approach of the CS, which comprises comprehensive management of OAC therapy as well as standardized monitoring of clinical status, complications, side effects and VKA-influencing factors (such as infections or nutrition) and registering these in an electronic patient file. On the basis of this information, algorithm-driven VKA dosing, patient education and counselling, individualized visits for control of anticoagulation therapy, supported self-measurement and management of OAC and telemedicine support are provided by specialized nurses and doctors [40].

## 5. Strengths and Limitations

A major strength of the present study is the large-scale investigation regarding the quality of OAC therapy predominantly performed with phenprocoumon in a real-life setting of RMC, and the evaluation of treatment improvement by a specialized CS. In this analysis, the established surrogate parameter TTR as well as prospective assessment of all relevant adverse events associated with VKA treatment in VTE patients were used to assess the quality of OAC. There are also limitations of this work that merit consideration, such as the use of a non-randomized study design might lead to potential bias. However, we performed two statistical approaches to address this potential source of bias, which provided consistent results, corroborating the robustness of the data interpretation. Nevertheless, in some cases, the limited sample size constituted another limitation, particularly in subgroup analyses and for the analysis of thromboembolic events. Detailed information on VTE (e.g., regarding etiology) was not available for analysis, but might be interesting to be incorporated in future studies on this topic. Due to the nature of an observational “real-world” study, selection and survival bias may limit the extrapolation of results.

## 6. Conclusions

This study underscores the benefits of a telemedicine-based coagulation service in comparison to regular medical care with respect to the quality of anticoagulation therapy and the clinical outcome of VTE patients receiving phenprocoumon. Importantly, the observed treatment benefits were independent of VTE indication, treatment duration and age, as well as the clinical profile. The findings may be particularly relevant for those patients in need of VTE prophylaxis who are not suitable for a DOAC-based treatment regime.

## Figures and Tables

**Figure 1 jcm-09-03281-f001:**
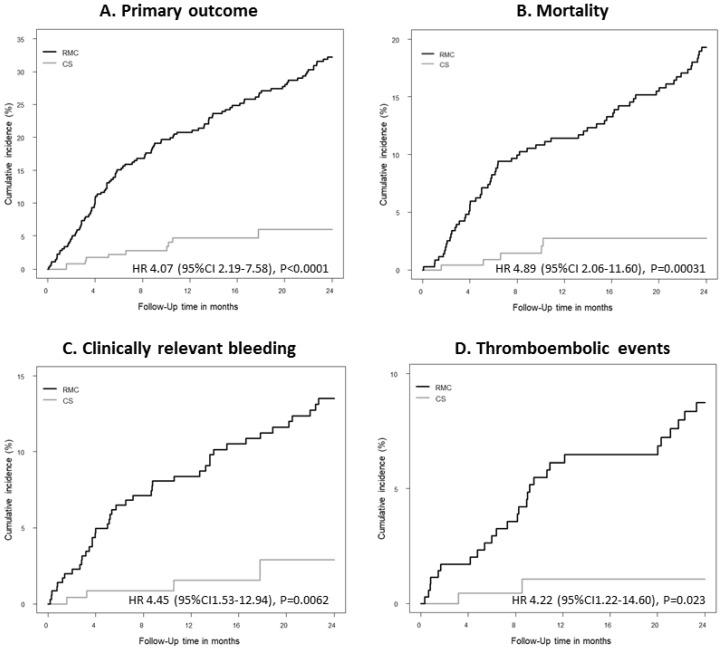
Propensity-weighted cumulative hazard plots of both groups. RMC in black color and CS in grey color for the adverse events under OAC treatment during the 2-year follow-up observational period (unadjusted propensity-weighted Cox regression models). (**A**) Primary outcome; (**B**) Mortality; (**C**) Clinically relevant bleeding; (**D**) Thromboembolic events.

**Figure 2 jcm-09-03281-f002:**
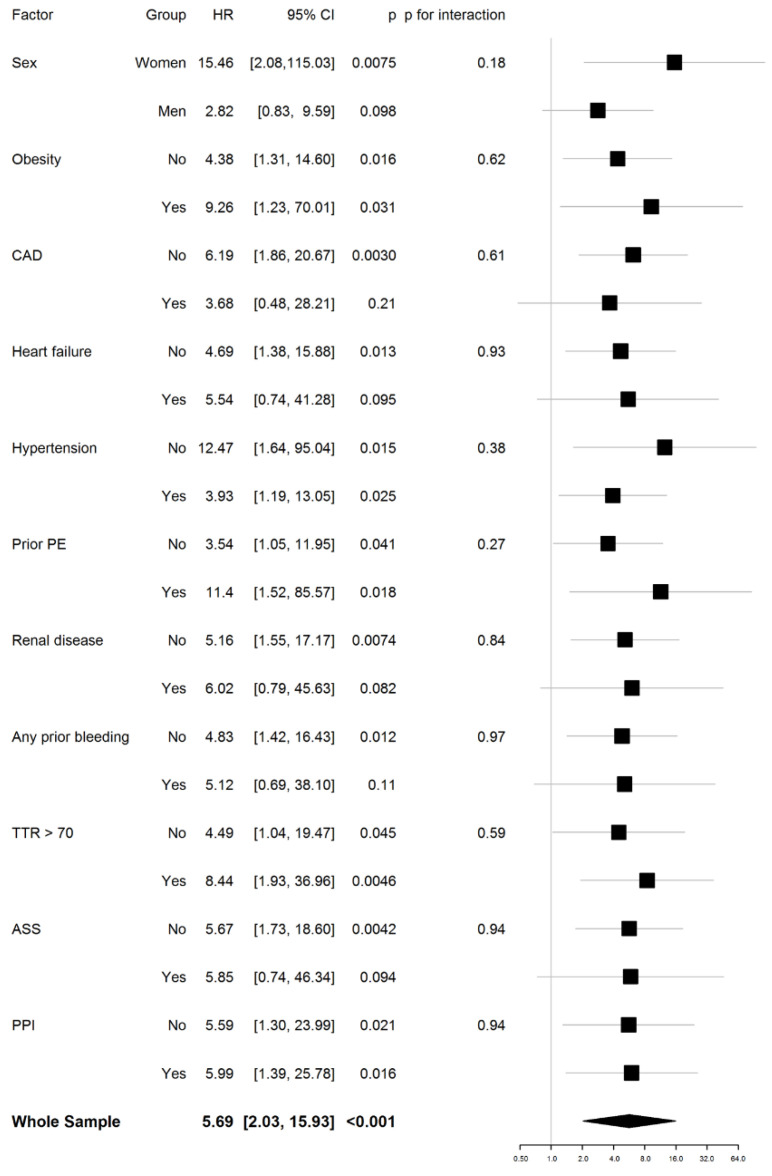
Relative risk of clinically relevant bleeding for patients in coagulation service in comparison to those in the regular medical care, according to subgroup (Wald test). ASS, aspirin; CAD, coronary artery disease; CI, confidence interval; HR, hazard ratio; PE, pulmonary embolism; PPI, proton pump inhibitor; TTR, time in therapeutic range (unadjusted Cox regression models).

**Figure 3 jcm-09-03281-f003:**
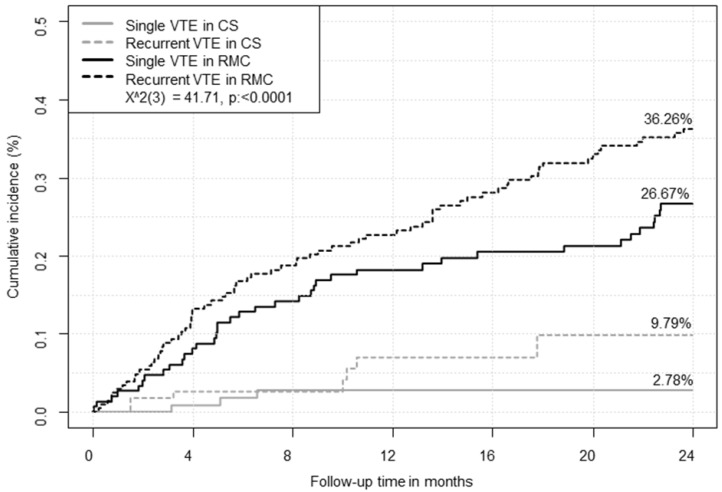
Comparison of net clinical benefit outcome of anticoagulated individuals with single and recurrent VTE events in regular medical care versus a telemedicine-based coagulation service. CS, coagulation service; RMC, regular medical care; VTE, venous thromboembolism.

**Figure 4 jcm-09-03281-f004:**
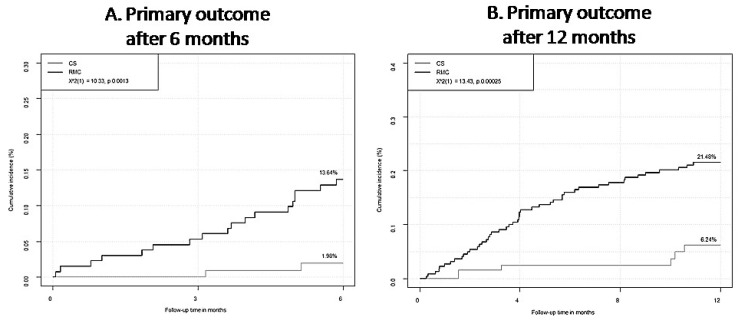
Comparison of net clinical benefit outcome in regular medical care (RMC) and coagulation service according to treatment duration of venous thromboembolism. (**A**) Primary outcome of venous thromboembolism (VTE) patients with single VTE event and without thrombophilia in their medical history after treatment duration of 6 months stratified by treatment modality (coagulation service (CS) vs. regular medical care (RMC)). *p* for difference between CS and RMC was analyzed by log-rank test. (**B**) Primary outcome of VTE patients with recurrent events or single events accompanied by existing thrombophilia in their medical history after treatment duration of 12 months stratified by treatment modality (CS vs. RMC). *p* for difference between CS and RMC was analyzed by log-rank test.

**Table 1 jcm-09-03281-t001:** Baseline characteristics of venous thromboembolism (VTE) patients in regular medical care (RMC) and coagulation service (CS).

Variable	Coagulation Service (*n* = 254)	Regular Medical Care (*n* = 360)	*p*-Value
Age (years; mean ± SD)	63.4 (±18.0)	68.3 (±14.5)	0.00035
Sex (Men)	44.9% (114/254)	53.9% (194/360)	0.033
CHA_2_DS_2_Vasc	3.73 (±2.02)	4.39 (±1.98)	0.00012
HAS-BLED	2.17 (±1.40)	2.69 (±1.36)	<0.0001
Charlson comorbidity index (mean ± SD)	4.42 (±2.71)	5.67 (±2.56)	<0.0001
**Sociodemographic Factors**			
Partnership	63.8% (162/254)	64.2% (231/360)	0.93
>2 Persons living in household	16.5% (42/254)	13.6% (49/360)	0.36
Children (at least one)	73.6% (187/254)	82.2% (296/360)	0.012
Nursing home inhabitants	6.7% (17/254)	4.7% (17/360)	0.37
Immigrants	5.1% (13/254)	10.8% (39/360)	0.012
Working	30.4% (77/253)	16.4% (59/360)	<0.0001
Higher School Certificate (Abitur)	18.1% (46/254)	15.3% (55/360)	0.38
School education (<10 years)	63.0% (160/254)	68.6% (247/360)	0.17
**Classical Cardiovascular Risk Factors**			
Arterial hypertension	59.8% (152/254)	71.9% (259/360)	0.0023
Diabetes	17.4% (44/253)	26.7% (96/360)	0.0082
Dyslipidemia	30.3% (77/254)	51.4% (185/360)	<0.0001
Family history of myocardial infarction or stroke	29.5% (75/254)	39.7% (143/360)	<0.0001
Obesity (BMI ≥ 30 kg/m^2^)	32.3% (82/254)	34.2% (123/360)	0.66
Smoking (currently)	7.1% (18/254)	10.6% (38/360)	0.16
**Concomitant Diseases**			
Atrial fibrillation	22.4% (57/254)	46.5% (166/357)	<0.0001
Cancer	22.9% (56/245)	20.1% (71/354)	0.42
Chronic lung disease	14.6% (37/254)	28.8% (102/354)	<0.0001
Coronary artery disease	16.7% (42/252)	37.5% (127/339)	<0.0001
Heart failure	13.8% (35/254)	36.1% (127/352)	<0.0001
History of myocardial infarction	9.1% (23/253)	21.8% (78/358)	<0.0001
History of stroke or transient ischemic attack	12.6% (32/254)	14.2% (51/359)	0.63
Liver disease	3.5% (9/254)	5.1% (18/356)	<0.0001
Peripheral artery disease	6.3% (16/254)	22.9% (79/345)	<0.0001
Renal disease	14.2% (36/254)	23.7% (85/358)	0.0038
Sleep apnea	4.8% (12/250)	9.8% (34/346)	0.029
Thrombophilia	12.6% (32/254)	9.2% (33/360)	0.18
**Quality of OAC therapy**			
TTR (median IQR)	76.9% (63.2–87.1%)	69.5% (52.3–85.6%)	<0.001
TuTR (median IQR)	6.4% (0.8–15.0%)	13.3% (2.2–27.9%)	<0.001
ToTR(median IQR)	10.6% (3.9–21.1%)	7.3% (0–21.7%)	0.033
Stable anticoagulation control	86.2% (207/240)	75.4(193/256)	0.0030
**Concomitant medication**			
Anti-platelet agents	19.7% (50/254)	18.1% (65/360)	0.67
Non-steroidal anti-inflammatory drugs	11.8% (30/254)	5.8% (21/360)	0.011
Proton pump inhibitor	38.6% (98/254)	35.0% (126/360)	0.39
Statin	24.4% (62/254)	32.5% (117/360)	0.031

Data are expressed as relative and absolute frequencies in binary variables. Normally distributed variables are shown as median with 25th/75th percentiles. Double entries were possible for those study participants in the CS cohort with prior treatment duration in RMC. Abbreviations: IQR indicates inter-quartile range; SD, standard deviation; CHA2DS2-VASc, CHA2DS2-VASc-Score (congestive heart failure—hypertension—age >75 years—diabetes mellitus—stroke/TIA—vascular disease—age 65–74 years—sex); HAS-BLED, HAS-BLED-score (hypertension—renal disease—liver disease—stroke history—prior major bleeding or predisposition to bleeding—labile international ratio [INR]); BMI, body-mass index; OAC, oral anticoagulation; TTR, time in therapeutic range; TuTR, time under therapeutic range; ToTR, time over therapeutic range.

**Table 2 jcm-09-03281-t002:** Comparison of adverse events in VTE patients between regular medical care (RMC) and coagulation service (CS).

	Events in RMC	Rate RMC	Events in CS	Rate CS	Rate Ratio (95% CI)	*p*-Value
Primary outcome	150	25.16	10	3.70	6.80 (3.60–14.47)	<0.001
All-cause mortality	65	10.90	5	1.85	5.89 (2.40–18.75)	<0.001
Clinically relevant bleeding	60	10.06	4	1.48	6.80 (2.52–25.76)	<0.001
Thromboembolic events	25	4.19	1	0.37	11.33 (1.85–465.26)	0.0015
Hospitalization	348	58.36	62	22.94	2.54 (1.94–3.39)	<0.001

Data are shown as absolute count of adverse events. Rate ratios (risk ratios) are calculated to compare the groups with *p* for difference. For analysis, all adverse events during the first 2 years after inclusion date were taken into account. All of the reported adverse events were under OAC treatment with inclusion of short bridging episodes or short treatment breaks (i.e., therapeutic breaks or bridging episodes). The events were reported as events per year per 100 patients. Bleeding events were categorized regarding to RELY criteria. Significant differences were seen if *p* was < 0.05. Abbreviations: RMC indicates regular medical care; CS, coagulation service; CI, confidence interval.

**Table 3 jcm-09-03281-t003:** Comparison of outcome in regular medical care and coagulation service by adjusted effect measures.

Variables of Adjustment	Adjustment for Age and Sex		Adjustment for Age, Sex and Charlson Index		Adjustment for Age, Sex Charlson Index and TTR	
	Hazard Ratio (95% CI)	*p*-Value	Hazard Ratio (95% CI)	*p*-Value	Hazard Ratio (95% CI)	*p*-Value
Primary outcome	5.39 (2.81–10.33)	<0.0001	5.04 (2.62–9.69)	<0.0001	5.01 (2.56–9.80)	<0.0001
All-cause mortality	5.54 (2.22–13.84)	<0.001	4.85 (1.94–12.15)	<0.001	4.77 (1.86–12.27)	0.0012
Clinically relevant bleeding	5.31 (1.89–14.89)	0.0015	5.38 (1.91–15.14)	0.0014	6.29 (2.20–17.96)	<0.001
Thromboembolic events	6.41 (1.51–27.24)	0.012	6.83 (1.60–29.12)	0.0094	6.31 (1.43–27.76)	0.015
Hospitalization	1.84 (1.34–2.55)	<0.001	1.76 (1.27–2.44)	<0.001	1.90 (1.35–2.68)	<0.001

Hazard ratios (HR) for the endpoints were presented with different adjustments. Hazard ratios of RMC are provided as multiples in comparison to the reference group of CS. Bleeding events were categorized regarding to RELY criteria. Significant differences were seen if *p* was < 0.05. Abbreviations: CI indicates confidence interval.

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
