# Peer review of "Telemedicine-Based Specialized Care Improves the Outcome of Anticoagulated Individuals with Venous Thromboembolism—Results from the thrombEVAL Study"

_jcm, 2020, doi:10.3390/jcm9103281_

Round 1

Reviewer 1 Report

this paper is ready for acceptance

Author Response

this paper is ready for acceptance

=> Response: Thank you very much for this positive feedback

Reviewer 2 Report

Thank you for submitting your paper.

I am sure it entailed a significant amount of work. Nevertheless I believe it needs a lot more work in order to be able to transfer the message you want.

I have found your references somewhat outdated - only 27% of them were published during the last 5 years.

There is an obvious need for an English native to review the text. There are innumerous grammar errors that preclude a good understanding of the message.

The text is very confusing, repetitive and very hard to read.

The methodology is also a bit confusing and difficult to understand. It is not completely clear as you add more variables and subgroup analysis in the middle of the text. It would need a major change to improve readability.

I will pin out only a few points in order to help you improve your text:

Page 2 -

4 - Computer - capital letter

Page 3 -

13 - IRQ should be presented as number - number not as number/number

13 - Throughout the text you use "()" in between "()". I believe you should use "[]" instead.

17 - "...RMC showed ..." this sentence is hard to understand

26 - Sentence needs improving

Page 5

22 - Please state what ACCP stands for

Page 7

12 - "...For the RMC cohort..." until the end of the page - very difficult to read, needs improving.

Page 8 -

1-4 - Very difficult to read, needs improving.

16 - This sentence is a bit out of context

Page 9

3 - 8 - Very difficult to read, needs improving.

18-19 - Very difficult to read, needs improving.

23-25 - Several grammar errors

Page 11

1 - "p" value is presented throughout the text as "P" and not as "p"

16 - "a priori" should be italic

19 - State more information about software

Page 12

14 - Why TTR was not possible to calculate for some patients?

16 -18 - Very difficult to read, needs improving.

Page 13

5 - "...TTR in self-managed patients..." - what do you mean?

13 - RMC and CS groups

14 - The groups were clinically different - I understand you tried to surpass this problem latter on but this needs to be explained here.

Page 14

5 - Please explain why you used a propensity score weighted analysis with unadjusted Cox regression.

12 - Table S2 shows a difference regarding peripheral artery disease

14 - Table S3 shows no difference for thromboembolic events.

Page 15

13 - "both" is being misused here

15 -19 - Very difficult to read, needs improving.

Page 16

5-11 - Repeat of results

22 - Improve sentence

24 - 25 - Repeat of results

Page 17

2 - 26 - This whole page needs a major improvement as it is very difficult to read and has innumerous grammar errors.

25 - What benefit?

Page 18

1 - Improve sentence

9 - 14 - Needs a major improvement as it is very difficult to read and has innumerous grammar errors.

22 - Please state what additional effects.

23 - "... treatment duration..." a repetition of what is written in the previous paragraph.

24 - 26 - Improve sentence

Page 19

1-4 - Very difficult to read, needs improving

14-15 - Improve sentence

17-20 - Repeated

Page 20

8 - ",both,"

14 - 18 - Improve sentences

Page 40

N from coagulation service - n=374.84 ?!

Author Response

Reviewer 2

I am sure it entailed a significant amount of work. Nevertheless I believe it needs a lot more work in order to be able to transfer the message you want.

  • Response: Thank you very much for this comment. We revised our manuscript and hope that you agree that the message is now clear and good to understand.

I have found your references somewhat outdated - only 27% of them were published during the last 5 years.

  • Response: As recommended, we included more references of papers, which were published in the last 5 years (≥2015). The proportion of publications ≥2015 was expanded to >45% of the references. Nevertheless, we need some of the older references to discuss our findings adequately.

There is an obvious need for an English native to review the text. There are innumerous grammar errors that preclude a good understanding of the message.

  • Response: Thank you very much for this coment. The manuscript was checked and revised by an English native speaker.

The text is very confusing, repetitive and very hard to read.

  • Response: We revised the whole manuscript in order to clarify the confusing parts.

The methodology is also a bit confusing and difficult to understand. It is not completely clear as you add more variables and subgroup analysis in the middle of the text. It would need a major change to improve readability.

  • Response: As recommended, we revised the description of the methodology. In particular, we included a specification of the subgroup analyses on page 11: “In addition, subgroups defined a priori were screened for potential interactions. Relative risk of clinically relevant bleeding as well as primary outcome for patients in CS in comparison to those in the RMC, according to different subgroups were analyzed with the Wald test.”

I will pin out only a few points in order to help you improve your text:

Page 2 -

4 - Computer - capital letter

  • Response: Thank you very much. We corrected this mistake.

Page 3 -

13 - IRQ should be presented as number - number not as number/number

  • Response: As recommended, we adapted this.

13 - Throughout the text you use "()" in between "()". I believe you should use "[]" instead.

  • Response: Thank you very much for this hint. We corrected this in the whole manuscript.

17 - "...RMC showed ..." this sentence is hard to understand.

  • Response: Thank you very much for this valuable comment. We clarified the statement: “Patients in RMC had higher risk for primary outcome (clinically relevant bleedings, thromboembolic events and mortality, HR 5.39 [95%CI 2.81-10.33],P<0.0001), mortality (HR 5.54 [2.22-13.84],P=0.00025), thromboembolic events (HR 6.41 [1.51-27.24],P=0.012), clinically relevant bleeding (HR 5.31 [1.89-14.89],P=0.0015) and hospitalization (HR1.84 [1.34-2.55],P=0.0002).”

26 - Sentence needs improving

  • Response: Thank you very much for this comment. We adapted the sentence: “CS was prognostically relevant, beyond providing advantages of improved INR monitoring.”

Page 5

22 - Please state what ACCP stands for

  • Response: Thank you very much for this hint. We included the information: ”Although a telemedicine-based, specialized coagulation service was recently reported to improve the outcome of patients receiving VKA [20-24], the current recommendations of the American College of Clinical Pharmacy (ACCP) do not support anticoagulation management services due to the heterogeneity of study results [25].”

Page 7 - 12 - "...For the RMC cohort..." until the end of the page - very difficult to read, needs improving.

Page 8 - 1-4 - Very difficult to read, needs improving.

  • Response: Thank you very much for this helpful comments. We revised the mentioned paragraph on pages 7/8: “Firstly, patients of the RMC cohort were enrolled during their hospitalization in 21 different hospitals. VTE Patients with an indication for anticoagulant treatment of at least 4 months duration were asked to participate in our observational study. Exclusion criteria comprised age <18 years, withdrawal of prior given consent, contraindications to VKA treatment (e.g. pregnancy or known hypersensitivity) or participation in other clinical trials. All patients who gave written informed consent, not fulfilling these exclusion criteria, were included in the RMC group without any further selection (Table S1 in the supplementary material). Secondly, regarding the CS cohort, VTE patients with an indication of at least 3 months of anticoagulant VKA treatment without any of the mentioned exclusion criteria were included in the CS group after giving written informed consent (Table S1 in the supplementary material). These patients, who were included in the CS, were on the one hand hospitalized or ambulatory treated patients at the University medical Center Mainz (Germany) or on the other hand patients, who were referred from their physicians to the CS for the management of the anticoagulant treatment. Therefore, the CS cohort comprises VTE patients, who were anticoagulation-naïve (new VTE diagnosis), as well as VTE patients, who were already under anticoagulation treatment, switching from RMC to the anticoagulant treatment managed by the CS.”

16 - This sentence is a bit out of context

  • Response: As suggested, we removed this sentence.

Page 9

3 - 8 - Very difficult to read, needs improving.

  • Response: Thank you very much for this helpful comment. We clarified these sentences: “We included an intra-individual comparison regarding the quality of oral anticoagulation of patients treated firstly in RMC and afterwards in coagulation service. To obtain this intra-individual comparison, data of patients, who were priorly treated in RMC and afterwards included in the CS, were separated in data regarding the treatment before inclusion in the CS and data of the period during the management in the CS [27]. The data, who belong to the period of the anticoagulant treatment before inclusion in the CS, were incorporated in the RMC cohort [27]. This approach enabled us to analyze an intra-individual comparison of these patients. Regarding the quality of anticoagulation control, INR values were obtained from anticoagulation documentation in RMC (e.g. from the anticoagulation pass) for a maximum period of 3 years and electronic patient files in the CS were queried for a maximum duration of 2 years.”

18-19 - Very difficult to read, needs improving.

  • Response: As suggested, we improved the sentence: “Systemic embolisms were defined as acute vascular occlusions of extremities or of any other organ.”

23-25 - Several grammar errors

  • Response: Thank you very much. We corrected the mistakes: “Major bleeding was defined as (i) a bleeding event with reduction in hemoglobin level of at least 2.0 g/L, or (ii) a bleeding event leading to transfusion of at least 2 units of blood or packed cells, or (iii) a symptomatic bleeding in a critical area or organ such as retroperitoneal bleeding, intra-articular bleeding or pericardial bleeding. Clinically relevant non-major bleeding was defined as a bleeding event requiring medical attention in either an ambulatory or clinical setting. Clinically relevant bleeding represents the composite endpoint of major bleeding and clinically relevant non-major bleeding.”

Page 11

1 - "p" value is presented throughout the text as "P" and not as "p"

  • Response: As suggested, we corrected this mistake.

16 - "a priori" should be italic

  • Response: As suggested, we adapted this.

19 - State more information about software

  • Response: As recommended, we included more information about the statistical software (page 11): “All statistical tests were two-sided. P values <0.05 were considered to indicate statistical significance. Statistical analyses were performed with R (https://www.R-project.org/; R Foundation for Statistical Computing, Vienna, Austria), version 3.4.3.”

Page 12

14 - Why TTR was not possible to calculate for some patients?

  • Response: In order to calculate patients’ TTR, at least a minimum of INR measurements during a timeframe is essential and needed. If this number of measurements is to low, it makes no sense to calculate a TTR.

16 -18 - Very difficult to read, needs improving.

  • Response: We revised these sentences: “While CS patients were more often in employments, RMC patients more frequently had children. Educational level, the rate of partnerships as well as the proportion of participants living in nursing homes were comparable between both groups.”

Page 13

5 - "...TTR in self-managed patients..." - what do you mean?

  • Response: Thank you very much for this comment. We clarified this sentence: “In contrast, TTR of patients, who measured their INR-values on their own, did not differ significantly between both groups (RMC: 88.2% [75.6%-95.3%] vs. CS: 87.4% [64.9%-97.6%], P=0.80).”

13 - RMC and CS groups

  • Response: As suggested, we adapted this sentence: “Mean follow-up times for RMC and CS were 19.9±7.4 and 12.8±7.3 months, respectively.”

14 - The groups were clinically different - I understand you tried to surpass this problem latter on but this needs to be explained here.

  • Response: Thank you very much for this valuable comment. We included the following sentences on page 14: “Since the patient characteristics of the RMC and CS groups were different (as shown in Table 1), we adjusted the Cox regression models for age and sex as well as age, sex and Charlson-Index, and furthermore for age, sex, Charlson-Index and TTR in order to test the widespread independence of these parameters. These adjusted Cox regression models revealed stable results regarding the benefit of CS on the mentioned adverse events (Table 3).”

Page 14

5 - Please explain why you used a propensity score weighted analysis with unadjusted Cox regression.

  • Response: Propensity score analysis is widely used in observational studies to adjust for confounding and estimate the causal effect of a treatment on the outcome. Since the propensity score weighting is used for baseline covariate adjustment, the Cox regression analyses were not further adjusted.

12 - Table S2 shows a difference regarding peripheral artery disease

  • Response: As shown in Table 1, the difference regarding the comorbidity peripheral artery disease between RMC and CS group was large. Thus, propensity score weighting was able to align both groups from a substantial difference regarding the comorbidity peripheral artery disease to a smaller difference, but this difference remained still slightly significant (Table S2 in the supplementary material).

14 - Table S3 shows no difference for thromboembolic events.

  • Response: Table S3 shows significant differences for the crude model, after adjustment with age and sex as well as age, sex and Charlson-Index. The competitive risk model was only in the fuly adjusted model with several scores not more significant.

Page 15

13 - "both" is being misused here

  • Response: Thank you very much. We corrected this: “Patients with single (HR 4.45 [95%CI 2.20-8.96], P<0.0001) as well as patients with recurrent (HR 5.45 [95%CI 2.79-10.63], P<0.0001) VTE events showed higher primary outcome rates in RMC compared to CS (adjusted for age, sex and Charlson-Index).”

15 -19 - Very difficult to read, needs improving.

  • Response: As recommended, we revised this paragraph: “VTE events showed higher primary outcome rates in RMC compared to CS (adjusted for age, sex and Charlson-Index). This finding was supported by the results of the Cox regression analysis demonstrating that management of OAC therapy in RMC was independently associated with higher incidence of adverse events regarding the primary outcome (HR 4.87 [95%CI 2.53-9.37], P<0.0001) compared to OAC management by the telemedicine-based CS (independent of age, sex, Charlson-Index and the differentiation between single and recurrent VTE).”

Page 16

5-11 - Repeat of results

  • Response: Thank you very much for this comment. According to the STROBE guideline, we included a paragraph early in the discussion section summarizing the main results of our paper.

22 - Improve sentence

  • Response: As suggested, we revised this sentence: “By contrast, other studies did not report similarly high percentages of TTR, reporting median TTR values below 70% [21,22,33].”

24 - 25 - Repeat of results

  • Response: Thank you very much for this comment, but in order to discuss our results, we have to explain our finding. Therefore, we decided to keep this short repetition of our results.

Page 17

2 - 26 - This whole page needs a major improvement as it is very difficult to read and has innumerous grammar errors.

  • Response: As suggested, we revised these paragraphs of the discussion.

25 - What benefit?

  • Response: Thank you very much for this helpful comment. We adapted the sentence: “We previously reported significantly improved clinical outcomes for the whole ThrombEval study sample, comprising all patients with various indications of OAC treatment [44].”

Page 18

1 - Improve sentence

  • Response:” Nevertheless, there are some general differences regarding OAC treatment of VTE patients in comparison to those patients with OAC treatment for AF or mechanical valves: in contrast to patients with OAC treatment due to AF or mechanical valves, whose OAC indications in the majority of cases are focused on primary prevention of ischemic stroke or valve complications, the treatment of VTE represents the combination of therapy for acute PE or DVT to solve thrombus or embolus and of secondary prevention to avoid recurrent VTE events. The fact that VKA management requires optimal anticoagulation quality to minimize risks including death due to recurrent PE provides a strong rationale for achieving optimal TTR in patients that are still treated with VKA [1-3,5,18,38,39,45,46].”

9 - 14 - Needs a major improvement as it is very difficult to read and has innumerous grammar errors.

  • Response: Thank you very much for this valuable comment. We revised this paragraph: “Remarkably, no significant interactions were detected in the a priori defined subgroups: In brief, we observed comparable treatment benefits due to CS in the subgroups of patients with renal insufficiency as well as in patients aged ≥80 years. This finding is of outstanding interest, since these patients might be candidates alternatively treated with reduced doses of DOACs.”

22 - Please state what additional effects.

  • Response: Thank you very much for this helpful comment. We included the following sentence in order to explain the ‘additional positive effects of CS’ : “These additional positive effects (beyond better dosing) of the CS might primarily attributed to better and closer supervision of the anticoagulated patients including better management of particular situations such as preparation for surgeries, perioperative OAC management or infections.”

23 - "... treatment duration..." a repetition of what is written in the previous paragraph.

  • Response: As recommended, we removed the part about treatment duration once.

24 - 26 - Improve sentence

  • Response: As suggested, we revised this sentence: “Notably, the achieved risk reductions regarding bleeding events as well as thromboembolic events of CS versus RMC were higher than the reported reductions achieved by DOAC treatment compared to warfarin in VTE patients [1,7,28,29].”

Page 19

1-4 - Very difficult to read, needs improving

  • Response: Thank you very much for your comment. We improved these sentences: “Nevertheless, it has to be mentioned that patients, who were enrolled for CS, might be more aware and be more motivated to achieve an optimized/better VKA management. Therefore, this difference in motivation and awareness might have resulted in a better adherence to the treatment recommendations, an improved TTR, and a lower incidence of adverse events and treatment complications in the patients of the CS.”

14-15 - Improve sentence

  • Response: Thank you very much. As suggested, we revised these sentences: “Although the concept of CS treatment was primarily developed for VKA treatment, our data suggest that CS might also be beneficial in OAC patients treated with DOACs. This beneficial and positive effect might be primarily explained by the supportive care of the CS in all OAC patients [3,24,50-52]. Thus, many CS in the United States have adapted their services to also provide care for patients taking DOACs [24,51].”

17-20 – Repeated

  • Response: As suggested, we removed this sentence.

Page 20

8 - ",both,"

  • Response: Thank you very much for this comment. We revised this sentence:” In this analysis, the established surrogate parameter TTR as well as prospective assessment of all relevant adverse events associated with VKA treatment in VTE patients were used to assess the quality of OAC.”

14 - 18 - Improve sentences

  • Response: As suggested, we revised this sentence: „Nevertheless, in some cases the limited sample size constituted another limitation, particularly in subgroup analyses and for the analysis of thromboembolic events.“

Page 40

N from coagulation service - n=374.84 ?!

  • Response: Thank you very much. We corrected this mistake.

Round 2

Reviewer 2 Report

Well done, you managed to improve substantially your manuscript. Now it is easily readable and I believe you managed to transfer you message with good result.

Thank you for following my advices.

There are still some minor points that need improvement:

Page 8

Line 9 - (Institute of Medical Biostatistics, Epidemiology and Informatics [IMBEI], ... [IZKS]...

Page 9

Line 5 - "The data, who belong to the period of the anticoagulant treatment before inclusion in the CS, were incorporated in the RMC cohort" - wrong grammar

Page 13

Line 5 - In contrast, TTR of patients who measured their INR-values on their own did not differ significantly between both groups...

Page 17

Line12 - ...The TTR obtained by CS in our study was significantly higher than in RMC (76.9% vs. 69.5%, P<0.001) and better than the median TTR values of control groups in larger registration trials comparing the use of DOACs in VTE patients... (would it be better?)

Page 20

Line 24 - "...eating characteristics..." this does not sound well.

Page 30

Line 7 - (Coagulation service [CS] vs. Regular Medical Care [RMC])

Line 12 - (Coagulation service [CS] vs. Regular Medical Care [RMC])

Page 32

Quality of OAC therapy - Please present IQR as "number - number" instead of "number/number"

Page 35

Line 7 - "P" not "p"

Page 36

Line 17 - "... suitable for a new OAC-based treatment regime"

Author Response

Reviewer 2

Well done, you managed to improve substantially your manuscript. Now it is easily readable and I believe you managed to transfer you message with good result.

  • Response: Thank you very much for this positive feedback.

Thank you for following my advices.

There are still some minor points that need improvement:

Page 8

Line 9 - (Institute of Medical Biostatistics, Epidemiology and Informatics [IMBEI], ... [IZKS]...

  • Response: Thank you very much for this comment. We adapted this: “Study monitoring was performed by independent institution (Interdisciplinary Center for Clinical Trials [IZKS], Germany).”

Page 9

Line 5 - "The data, who belong to the period of the anticoagulant treatment before inclusion in the CS, were incorporated in the RMC cohort" - wrong grammar

  • Response: We corrected the grammar: “The data, which belong to the period of the anticoagulant treatment before inclusion in the CS, were incorporated in the RMC cohort [27].”

Page 13

Line 5 - In contrast, TTR of patients who measured their INR-values on their own did not differ significantly between both groups...

  • Response: Thank you very much. We corrected this mistake.

Page 17

Line12 - ...The TTR obtained by CS in our study was significantly higher than in RMC (76.9% vs. 69.5%, P<0.001) and better than the median TTR values of control groups in larger registration trials comparing the use of DOACs in VTE patients... (would it be better?)

  • Response: As suggested, we revised this sentence: “The TTR obtained by CS in our study was significantly higher than in RMC (76.9% vs. 69.5%, P<0.001) and higher than the median TTR values of the control groups in the large registration trials for the DOACs in VTE patients [1,7,28,29]”

Page 20

Line 24 - "...eating characteristics..." this does not sound well.

  • Response: Thank you very much for this comment. We corrected this: “These benefits may be explained by the multi-factorial approach of the CS, which comprises comprehensive management of OAC therapy as well as standardized monitoring of clinical status, complications, side effects and VKA-influencing factors (such as infections or nutrition) and registering these in an electronic patient file. “

Page 30

Line 7 - (Coagulation service [CS] vs. Regular Medical Care [RMC])

Line 12 - (Coagulation service [CS] vs. Regular Medical Care [RMC])

  • Response: As recommended, we changed the brackets: “A: Primary outcome of venous thromboembolism (VTE) patients with single VTE event and without thrombophilia in their medical history after treatment duration of 6 months stratified by treatment modality (Coagulation service [CS] vs. Regular Medical Care [RMC]). P for difference between CS and RMC was analyzed by log-rank test. B: Primary outcome of VTE patients with recurrent events or single events accompanied by existing thrombophilia in their medical history after treatment duration of 12 months stratified by treatment modality (Coagulation service [CS] vs. Regular Medical Care [RMC]). P for difference between CS and RMC was analyzed by log-rank test.”

Page 32

Quality of OAC therapy - Please present IQR as "number - number" instead of "number/number"

  • Response: As suggested, we adapted this.

Page 35

Line 7 - "P" not "p"

  • Response: We adapted this: “Significant differences were seen if P was <0.05.”

Page 36

Line 17 - "... suitable for a new OAC-based treatment regime"

  • Response: Thank you very much for this helpful comment. We corrected this mistake: “The findings are relevant for patients necessitating VTE prophylaxis, who are not suitable for a new OAC-based treatment regime.”